# Influence of Different Single Factors on the Spatial-Temporal Distribution Law of Phosphorus in the Generalized River

**Pengjie Hu** [1,2,†], **Jin Xu** [1,2,†], **Lingling Wang** [1,2,*], **Hongwu Tang** [1,2], **Mengtian Wu** [1,2] **and Pengcheng Xu** [1,2]

1   State Key Laboratory of Hydrology-Water Resources and Hydraulic Engineering, Nanjing 210098, China; hpj@hhu.edu.cn (P.H.); hhu_xj@hhu.edu.cn (J.X.); hwtang@hhu.edu.cn (H.T.); wmtsky@hhu.edu.cn (M.W.); xupengcheng@hhu.edu.cn (P.X.)
2   College of Water Conservancy and Hydropower Engineering, Hohai University, Nanjing 210098, China
*   Correspondence: wanglingling@hhu.edu.cn
†   These authors contributed equally to this work.

**Abstract:** Phosphorus is the main limiting factor in river eutrophication, whose distribution law is affected by sediment and hydrodynamics. Based on the sediment of Wujiadu Hydrological Station in the middle reaches of the Huaihe River, a physical experiment of phosphorus adsorption onto sediment in a dynamic environment was carried out in this paper to obtain two important parameters of the water quality model. By considering the effect of adsorption and desorption, diffusion, sedimentation and suspension, a numerical model of a three-dimensional periodic flume was established, verified and then applied to the generalized river according to the shape of the cross section in the Bengbu reach of the Huaihe River. The spatial-temporal distribution law of phosphorus was obtained with different single inflow factors. When $DP$ or $AP$ changed, the overall trend of downstream $DP$ was consistent with that of incoming $DP/AP$; however, it became more complex when $SS$ or $Q$ changed, which aggravated the sedimentation and re-suspension process. When incoming $DP$ changed, $DP_{max}$ decreased with the distance, while when incoming $AP$ or $SS$ changed, $DP_{max}$ increased at first and then decreased. The relationship between $\Delta DP_{max}$ and $\Delta TP_{in}$ was fitted into a straight line with a slope of 0.9951, while it was 0.2154 with the change of incoming $SS$. The effect of the not-constant time of the inflow on the peak concentration of phosphorus along the river was much weaker than that of the peak concentration.

**Keywords:** generalized river; phosphorus; spatial-temporal distribution; different single factors

## 1. Introduction

Water pollution incidents occur frequently in rivers, among which the problem of eutrophication is more significant [1]. Compared with nitrogen, phosphorus is a more important limiting factor [2,3]. Sediment has a strong effect of adsorption and desorption on phosphorus [4,5], and the hydrodynamics will also change the distribution of phosphorus between the overlying water layer and the bed sediment layer [6,7]. With a large specific area [8] and high flocculation degree [9], the smaller the sediment particle size, the greater the phosphorus adsorption capacity. There is a positive correlation between the adsorption capacity of sediment for phosphorus and the iron, aluminum oxide and hydroxide on the surface [10,11]. The increase in velocity will enhance the disturbance of the water–sediment interface, promote the exchange of overlying water and pore water [12] and aggravate the transformation of adsorbed phosphorus between the suspended sediment and the bed sediment [13]. With the increase in sediment concentration, the absolute amount of dissolved phosphorus in water increases, but the amount of sediment adsorption per unit mass decreases [14]. When the initial phosphorus concentration increases, the equilibrium concentration of phosphorus absorbed by unit mass sediment increases, with longer time to reach equilibrium state [3,15].

It is already well-known that both hydrodynamics and sediment greatly affect the adsorption and desorption process of phosphorus. However, most of these studies are the results of the full reaction between sediment and phosphorus on a laboratory flume scale. In natural rivers, there is a dynamic balance between sediment and phosphorus because of inadequate reactions [16,17]. Based on the analysis of the measured suspended sediment and total phosphorus concentration in the Jialing River of the Yangtze River, it was found that there is a linear relationship between them, with obvious characteristics of seasonal variation [18], which can also be obtained from the research on the river of Hohhot section of the Yellow River [19] and Hangzhou Bay in the Yangtze River estuary [20]. According to the field experiment based on the Shaying River, there was an obvious positive relationship between the gate opening and the phosphorus concentration downstream of the gate, mainly due to the increase in the bottom disturbance of the incoming water and the increase in the incoming concentration [21]. The distribution of phosphorus above and below the sluice can also be affected by the sluice or dam. Over 70% of the sediment load and more than half of the phosphorous load were intercepted by deposition in the Three Gorges reservoir [22]. More than half of the rivers flowing to the sea will experience greater removal of silicon over nitrogen and phosphorus, impacting the role of diatoms in nearshore marine production [23].

Huaihe River, the third largest river in China, is also subjected to severe eutrophication and cyanobacterial blooms as a result of development and urbanization. A water quality model [24,25] containing Huaihe River and its two tributaries was carried out to analyze the distribution law under the action of a sewage leakage incident. However, the role of sediment was considered only in a generalized way. Furthermore, the distribution law obtained was quite different, mainly because the hydrodynamics caused by sediment and river shape vary in different places, and the parameters obtained by mathematical models were also quite different. In order to obtain the spatial-temporal distribution law under different incoming conditions based on the Huaihe River, this paper carried out a physical experiment of phosphorus adsorption onto sediment sampled from Wujiadu Hydrological Station to obtain two important parameters of the water quality model. By taking into account the effect of adsorption and desorption, diffusion, sedimentation and suspension, a numerical model of a three-dimensional periodic flume was established, verified and then applied to the generalized river according to the shape of the cross section in the Bengbu reach of the Huaihe River. The spatial-temporal distribution law of phosphorus was obtained with different single inflow factors.

## 2. Three-Dimensional Periodic Model

### 2.1. Concenpt and Govering Equations

In our previous work [26,27], the interaction law among hydrodynamics, sediment and phosphorus was obtained by the dynamic water experiment based on the sediment sampled at Wujiadu Hydrological Station of the Huaihe River. Two important parameters in the water quality model were obtained by fitting analysis. On this basis, considering the effect of the bed sediment, a hydrodynamic–sediment–phosphorus numerical model was established and verified, which took into account the adsorption and desorption between the dissolved and adsorbed phosphorus, the diffusion between the interface of overlying layer and bed layer, the sedimentation and the suspension of sediment. The hydrodynamic module and sediment–phosphorus module are described below.

The hydrodynamic module is established based on the continuity Equation (1) and the momentum Equation (2). In the physical experiment, the annular flume is driven by an external motor with plexiglass gears. In the numerical model, the periodic boundary is used, and the constant and uniform flow is obtained by increasing the horizontal volume force:

$$\nabla \cdot \boldsymbol{u} = 0 \tag{1}$$

$$\frac{\partial \boldsymbol{u}}{\partial t} + (\boldsymbol{u} \cdot \nabla)\boldsymbol{u} = -\frac{1}{\rho}\nabla p + \nu \cdot \nabla^2 \boldsymbol{u} + \boldsymbol{f} \tag{2}$$

where $u$ is the velocity vector; $\nabla$ is the Laplace operator; $t$ is the time; $p$ is the pressure; $\nu$ is the kinematic viscosity, $10^{-6}$ m$^2$/s at the normal temperature of 20 °C; and $f$ is the body force, which is equal to the resultant force of the gravitational acceleration in the vertical direction and the stream body force. In order to obtain a constant and uniform water flow in the numerical study, a streamwise body force, $F_x$, is added to counteract the kinetic energy loss caused by solid boundary resistance and water viscosity.

There is a very complex interaction mechanism between sediment and phosphorus, as shown in Figure 1. Phosphorus exists in two forms, namely, adsorbed phosphorus and dissolved phosphorus, including *DP* (dissolved phosphorus) and *AP* (adsorbed phosphorus) in the overlying water layer and *DPS* (dissolved phosphorus in bed sediment layer) and *APS* (adsorbed phosphorus in bed sediment layer) in the bed sediment layer. Sediment also exists in two forms, namely, *SS* (suspended sediment) and *BS* (bed sediment). There are several major functions in the diagram, namely, adsorption and desorption, which can be considered by the improved Langmuir model; diffusion, which can be considered by Fick's law; and sedimentation and resuspension, which can be considered by Stokes sedimentation rate formula and resuspension rate, respectively.

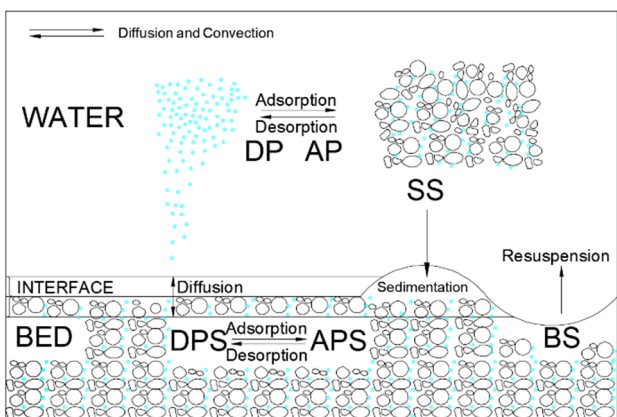

**Figure 1.** Conceptual model of interaction between phosphorus and sediment.

On the one hand, the dissolved phosphorus *DP* in the overlying water is adsorbed by the suspended sediment *SS*; on the other hand, it diffuses with the dissolved phosphorus *DPS* in the pore water, and its changing process over time is shown in Equation (3). On the one hand, the adsorbed phosphorus *AP* in the overlying water will be desorbed into the overlying water; on the other hand, it will enter the bed sediment layer with the sedimentation of the suspended sediment, and its changing process over time can be seen in Equation (4). *DPS* dissolved in the pore water of the bed sediment layer and *APS* adsorbed on the bed sediment can be adsorbed and desorbed. There will also be diffusion between *DPS* and *DP*, as specified in Equation (5). *APS* will enter the overlying water layer with the upward suspension of *BS* in the bed sediment layer, shown in Equation (6). The suspended sediment *SS* will settle into the upper layer of the bed, and the bed sediment *BS* will enter the overlying water layer under the action of shear force, which can be seen in Equations (7) and (8), respectively. This paper focuses on the analysis of the adsorption and desorption of sediment, so both the suspended sediment and bed sediment are regarded as the concept of concentration sediment, and their adsorption and desorption of phosphorus, as well as their own sedimentation and re-suspension are considered, but the specific starting process of sediment under the action of current is not taken into account, nor are the specific shape changes of the river bed.

$$\frac{\mathrm{d}DP}{\mathrm{d}t} = [k_d \cdot AP - k_a \cdot DP \cdot (B_m - AP)] \cdot SS - D \cdot \frac{2 \cdot (DP - DPS)}{H \cdot H_u} \tag{3}$$

$$\frac{\mathrm{d}AP}{\mathrm{d}t} = k_a \cdot DP \cdot (B_m - AP) - k_d \cdot AP - \frac{srt \cdot AP}{H_u} + rrt \cdot \frac{APS}{H_u \cdot SS} \cdot 10^{-3} \tag{4}$$

$$\frac{\mathrm{d}DPS}{\mathrm{d}t} = [k_{ds} \cdot APS - k_{as} \cdot DPS \cdot (B_m - APS)] \cdot \frac{BS}{H_d} \cdot 10^{-3} + D \cdot \frac{2 \cdot (DP - DPS)}{H \cdot H_d} \cdot 10^{-3} \tag{5}$$

$$\frac{\mathrm{d}APS}{\mathrm{d}t} = k_{as} \cdot DPS \cdot (B_m - APS) - k_{ds} \cdot APS + \frac{srt \cdot SS \cdot AP}{BS} \cdot 10^3 - rrt \cdot \frac{APS}{BS} \tag{6}$$

$$\frac{\mathrm{d}SS}{\mathrm{d}t} = -\frac{srt \cdot SS}{H_u} + \frac{rrt}{H_u} \cdot 10^{-3} \tag{7}$$

$$\frac{\mathrm{d}BS}{\mathrm{d}t} = srt \cdot SS \cdot 10^3 - rrt \tag{8}$$

where $t$ is the time; $DP$ is the dissolved phosphorus in overlying water, mg/L; $AP$ is the adsorbed phosphorus in overlying water, mg/g; $DPS$ is dissolved phosphorus in pore water of bed sediment, mg/L; $APS$ is adsorbed phosphorus on bed sediment, mg/g; $SS$ is suspended sediment, g/L; $BS$ is bed sediment, g/m²; $k_a$ is the adsorption coefficient of overlying water layer; $B_m$ is the maximum adsorption amount of sediment per unit mass, mg/g; $k_d$ is the desorption coefficient of the overlying water layer; $k_{as}$ is the desorption coefficient of the bed sediment layer; $k_{ds}$ is the desorption coefficient of the bed sediment layer; $H$ is the total thickness of the overlying water layer and bed sediment layer, m; $H_u$ is the water depth of the overlying water layer, m; $H_d$ is the depth of bed sediment layer, m; $D$ is the diffusion coefficient between the $DP$ of dissolved phosphorus in the overlying layer and the $DPS$ of dissolved phosphorus in pore water; $srt$ is the settling velocity of sediment, m/s; and $rrt$ is the resuspension rate of bed sediment, unit g/m²/d.

The above six variables are added to the Navier–Stokes equation in the form of a source term, as shown in Equation (9):

$$\frac{\partial X}{\partial t} + u\frac{\partial X}{\partial x} + v\frac{\partial X}{\partial y} + w\frac{\partial X}{\partial z} = D_x\frac{\partial^2 X}{\partial x^2} + D_y\frac{\partial^2 X}{\partial y^2} + D_z\frac{\partial^2 X}{\partial z^2} + S' \tag{9}$$

where $X$ is the six variables mentioned above, namely, $DP$, $AP$, $DPS$, $APS$, $SS$ and $BS$; $t$ is the time; $x$, $y$ and $z$ are the coordinate directions; $u$, $v$ and $w$ are the speed of directions $x$, $y$ and $z$, respectively; $D_x$, $D_y$ and $D_z$ are the diffusion coefficients of directions $x$, $y$ and $z$, respectively; and $S'$ is the source term corresponding to the six variables.

*2.2. Model Verification*

In order to further verify the applicability of the parameter formula under the combined action of suspended sediment and bed sediment, this section established and verified the corresponding numerical model on the basis of physical experiment of Haoke Cheng [28].

As shown in Figure 2, this section established a three-dimensional periodic circulating flume with the same size as the straight section of the annular flume in the physical experiment, that is, 0.21 m × 0.07 m × 0.07 m. The volume force $F_x$ along the flow direction was added to counteract the kinetic energy loss caused by solid boundary resistance and water viscosity, and a constant and uniform flow was obtained. The computational grid resolution is 60 × 20 × 20 (L × W × H), which is enough to obtain a stable numerical solution. The initial condition of the numerical simulation is the velocity field after the steady-state of the flow in the physical experiment. The initial water level is 0.07 m in all different cases.

In order to verify the hydrodynamic module, the case in which the average velocity of the cross section is 0.31 m/s were compared. As shown in Figure 3, the black solid frame is the time-averaged velocity of the numerical flume straight line AB (Figure 2) along the water depth direction, and the white hollow box is the measured velocity of 2.4 cm above the water–sediment interface of the straight section in the physical experiment, which showed that the two were in good agreement.

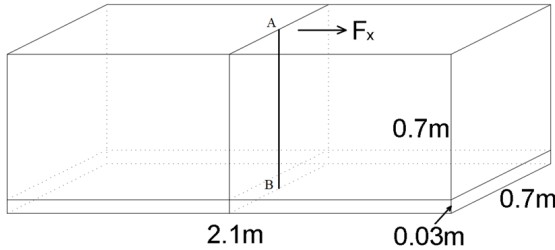

**Figure 2.** Periodic circulating flume.

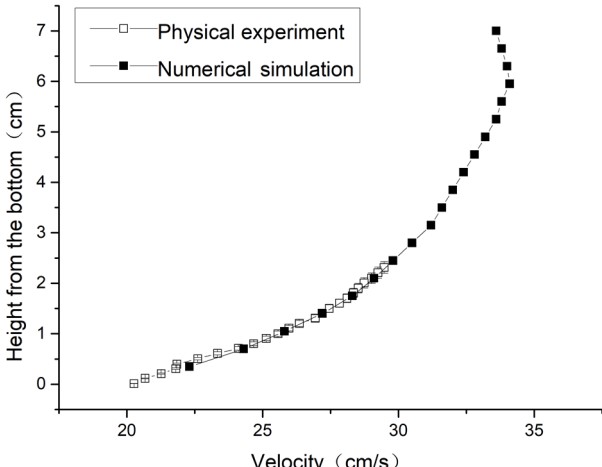

**Figure 3.** Comparison between numerical simulation and physical experiment on the changing process of flow velocity with water depth at AB, where the average velocity of cross section is 0.31 m/s.

The physical experiment adopts the design of full-section sand laying, with a sediment weight of 400 g, a moisture content of 33%, a sediment thickness of 3 mm and a sediment laying area of 1026 cm². It was calculated that the area density of bed sediment *BS* is 3898.6 g/m², but due to the partial saturated–unsaturated adsorption of phosphorus, the numerical simulation assumed that all bed sediments can absorb saturated phosphorus. This will make the sediment adsorb more phosphorus in the numerical simulation than in the physical experiment, so the reduction coefficient $k_h$ was set between 0 and 1. The *DP* of dissolved phosphorus in overlying water is 5 mg/L, and the concentration of *SS* in suspended sediment is 0.722 g/L after the preliminary experiment. The values of dissolved phosphorus and adsorbed phosphorus in the bed sediment layer were set to a small non-zero value, as shown in Table 1.

**Table 1.** Initial value of each variable of verifying case A.

| *DP* | *AP* | *DPS* | *APS* | *SS* | *BS* | *v* |
|------|------|-------|-------|------|------|-----|
| (mg/L) | (mg/g) | (mg/L) | (mg/g) | (g/L) | (g/m²) | (m/s) |
| 5 | 0.001 | 0.01 | 0.001 | 0.722 | 3898.6 | 0.315 |

Figure 4 showed the changing process of the ratio of total phosphorus to initial phosphorus concentration over time between the numerical simulation and physical experiment, which were in good agreement with each other. The numerical simulation also showed the concentration of total phosphorus in equilibrium state, 3 mg/L, with *DP* 2.91 mg and *AP* 0.12 mg/g. The bed sediment layer adsorbed 40% of the initial total phosphorus, and the reduction coefficient $k_h$ was 0.5.

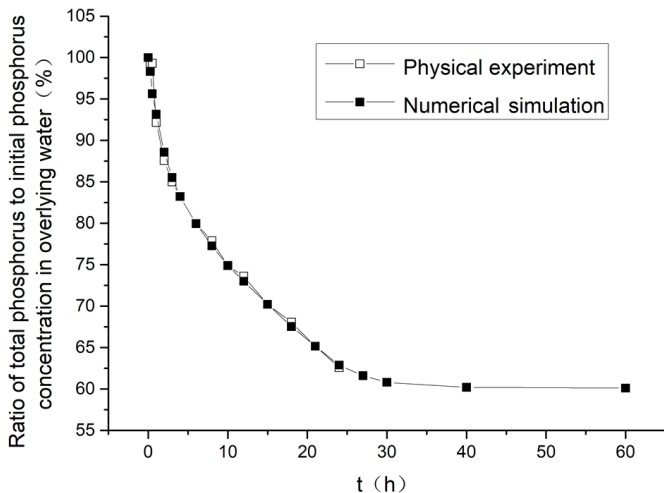

**Figure 4.** Comparison between numerical simulation and physical experiment on the changing process of ratio of total phosphorus in overlying water to initial phosphorus concentration.

## 3. Numerical Generalized River

### 3.1. Condition of Equilibrium Case

According to the measured cross-section and topographic data of the middle reaches of the Huaihe River, the generalized channel is 500 m wide and 5 m deep. Because the analysis focuses on the spatial-temporal distribution of phosphorus and sediment, a rectangular channel was directly adopted instead of a compound channel. At the same time, in order to take into account the reaction process of adsorption and desorption as much as possible, combined with the influence of flow velocity, a long straight channel was designed, which was 60 km. There were two bases for selecting this length. One was the influence scope of pollution events. Although the pollution length of the Huaihe River accident reached 130 km in July 1994 [29] and 150 km in July 2004 [30], the pollution near the lower reaches was relatively small, and the influence lengths of general pollution events were about 60 km. Another base was the reaction time. The indoor flume experiment showed that the adsorption and desorption process of sediment and phosphorus reached equilibrium after about 30 h. In order to analyze the difference between convection diffusion and adsorption desorption, the time of water flow from the upstream boundary to the downstream boundary at the maximum flow rate in this paper should be more than 30 h.

According to the measured sediment samples in the middle reaches of the Huaihe River and other literatures, it is generally considered that the area around the first 10 cm on the surface of the bed sediment is closely exchanged with the overlying water, and the reduction coefficient $k_h$, which counteracts the unsaturated adsorption, takes the value determined by the above rate. The calculation area of the generalized model is shown in Figure 5. The upper layer is a 5 m water layer, and the lower layer is the bed sand layer.

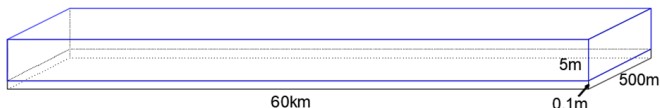

**Figure 5.** Schematic diagram of the generalized river (not scaled).

This paper mainly studied the influence of changing boundary conditions on the temporal and spatial distribution of phosphorus, so it is necessary to eliminate the influence of adsorption and desorption between non-equilibrium sediment and phosphorus in the initial field. Therefore, a water quality field in equilibrium, used as the initial field in the following simulation area, was obtained in this section.

In order to obtain the water quality field in equilibrium, a periodic boundary flume with the cross-section shape of the flume shown in Figure 5 was set. The average velocity 0.3 m/s of a cross-section in the physical experiment was selected as the calculation model condition, the water depth is 5 m and a stable hydrodynamic field is obtained.

The Surface Water Environmental Quality Standard GB3838-2002 [31] issued by the State Environmental Protection Administration stipulates that the total phosphorus 0.2 mg/L of overlying water in a river is the third class of water quality. Because the content of suspended sediment in Huaihe River is low and the ratio of phosphorus absorbed by the suspended sediment to the total phosphorus in the overlying water is low, for the convenience of numerical value, the dissolved phosphorus in overlying water $DP$ was determined to be 0.2 mg/L. According to the measured suspended sediment data in the middle reaches of the Huaihe River, the suspended sediment concentration $SS$ was selected as 0.5 g/l, with a median sediment particle size $D_{50}$ of 14.728 μm, a porosity of 0.45 and a water content of bed sediment of 33%. According to the Stokes settling velocity formula, the settling velocity is calculated to be 0.0142 cm/s, that is, 12.3 m/d. The bed sediment suspension rate $rrs$ refers to the results of Kaiming Hu [32]. When the average velocity of the cross section is 0.3 m/s, it is 6150 g/m$^2$/d.

On the basis of this constant hydrodynamic field, the concentration of phosphorus adsorbed by suspended sediment ($AP$) was continuously adjusted, and finally, a set of basic equilibrium case B, that is, the equilibrium state of $DP$, $AP$ and $SS$ will not be changed over time, was obtained. The values of the variables in the equilibrium case B are shown in Table 2, which was taken as the initial field for the whole area of the generalized river.

**Table 2.** Result value of each variable of equilibrium case B.

| $DP$ | $AP$ | $DPS$ | $APS$ | $SS$ | $BS$ | $v$ |
|---|---|---|---|---|---|---|
| (mg/L) | (mg/g) | (mg/L) | (mg/g) | (g/L) | (g/m$^2$) | (m/s) |
| 0.2 | 0.025 | 0.2 | 0.025 | 0.5 | 147,000 | 0.3 |

*3.2. Case Setting*

The Surface Water Environmental Quality Standard GB3838-2002 stipulates that five kinds of the total phosphorus in the overlying water are 0.02, 0.1, 0.2, 0.3 and 0.4 mg/L, respectively. Because the content of suspended sediment in Huaihe River is low, and the ratio of phosphorus absorbed by suspended sediment to total phosphorus in overlying water is low, for the convenience of numerical value, the peak values of dissolved phosphorus $DP$ in overlying water were set to be 0.02, 0.1, 0.3 and 0.4 mg/L, respectively, and then a higher concentration 1 mg/L was added. As shown in Figure 6, each case increases or decreases from the same value as the initial condition to the five peak concentrations listed above, then linearly decreases or increases to the same value as the initial condition and then keeps the boundary condition value unchanged until the end of the calculation. In order to analyze the influence of a single factor on the temporal and spatial distribution of phosphorus, the values of other incoming boundary conditions are set to the same value as the result of equilibrium case B. The not-constant time of the whole unsteady process is 36 h, and that before and after the peak is 18 h. In order to make the position of each section in the calculation area finally reach the near-equilibrium state, the total calculation time of different cases is also different, so the process of unsteady phosphorous and the partially constant process of dissolved phosphorus $DP$ over time are mainly listed in Figure 6. The changing processes of other cases over time are similar.

A group of cases with serious water pollution are selected, with incoming $DP$ 0.4 mg/L and the time of not-constant process $T_{nc}$ 12, 24, 36, 48 and 60 h. The specific case setting is shown in Figure 6. For the other three variables with peak changing, $AP$, $SS$ and $Q$ are set to 0.005/0.01/0.04/0.08/0.12 mg/g, 0.1/0.25/1/2/5 g/L and 350/600/875/1000/1250 m$^3$/s, respectively.

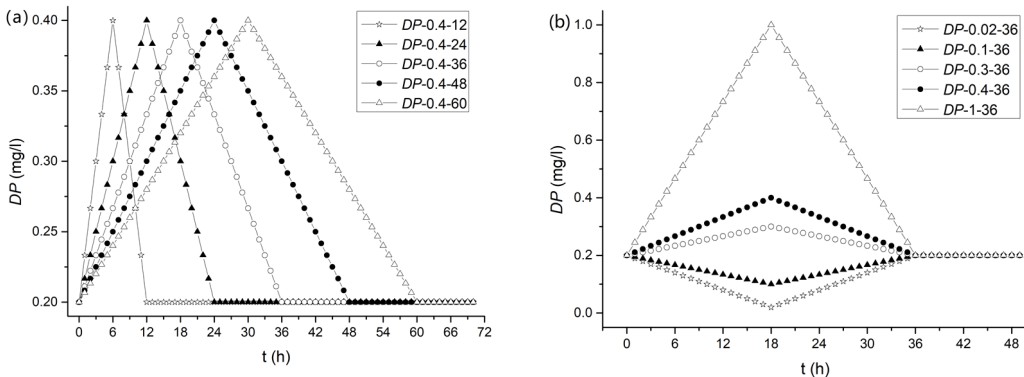

**Figure 6.** *DP* with different peaks or times of incoming flow. (**a**)incoming *DP* 0.4 mg/L, (**b**)the other three variables with peak changing.

## 4. Results and Discussions

### 4.1. Spatial-Temporal Distirbution Law

In order to explore whether phosphorus concentration, sediment concentration and hydrodynamic force have obvious environmental effects on the temporal and spatial distribution of phosphorus, this section analyzed the spatial-temporal distribution of phosphorus and sediment along the reach when different types of individual inflow factors changed.

Figure 7 shows the changing process of *DP* concentration of case *DP*-0.3-36 over time. For a better understanding, the case name in this paper is named as X-X-X, where the first X is the change of specific factor, the second X is the peak of the not-constant process of this factor and the third X is the duration of the not-constant process of this factor. It can be seen that when the concentration of the upstream boundary is transferred to different locations due to convection and diffusion, the *DP* of this location begins to change, which further proves that the change in phosphorus in the downstream is only due to the change in the upper boundary, instead of the initial condition, and the time of this stage is defined as $T_0$. Because the incoming *DP* has an upward unsteady process while other boundary conditions such as dissolved phosphorus and suspended sediment do not change, there will be an upward process of *DP* along the downstream; however, the peak concentration at different locations decreases with the increase in distance, mainly because the suspended sediment and bed sediment in the downstream will absorb the dissolved phosphorus in the water, and the time of this stage is defined as $T_1$. The changing trend of *DP* of other cases, that is, *DP*-0.02/0.1/0.4/1-36, is consistent with that of Figure 7.

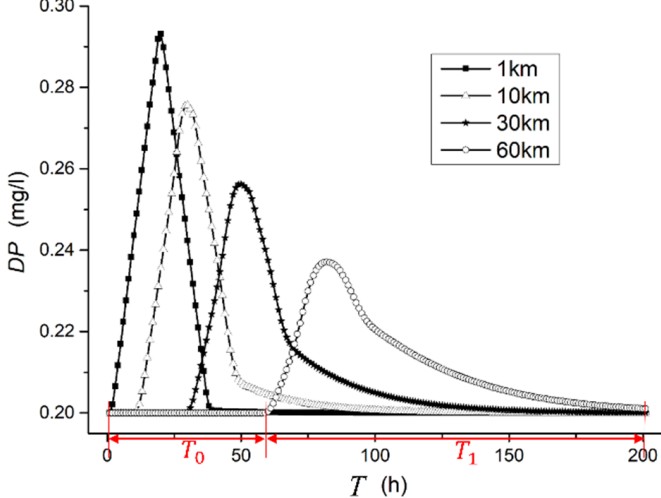

**Figure 7.** The changing process of *DP* concentration of case *DP*-0.3-36 over time.

Figure 8 shows the changing process of *DP* concentration of case *AP*-0.04-36 at the $T_1$ stage. It can be seen that when the *AP* of the incoming flow changes, the overall shape of the downstream *DP* over time is similar to that shown in Figure 7, but the maximum peak does not appear in the first cross-section. The adsorbed phosphorus *AP* affects the dissolved phosphorus *DP* through sediment desorption, and the action time is obviously longer than the convection diffusion time, while the incoming *DP* does not change, so the first several sections are affected by the *DP*, which will weaken the effect of *AP* on the *DP* at this position. Therefore, when the incoming flow *AP* changes, the peak value of downstream *DP* does not appear in the first section. The changing trend of the *DP* of other cases, that is, *AP*-0.005/0.01/0.08/0.12-36, is consistent with that of Figure 8.

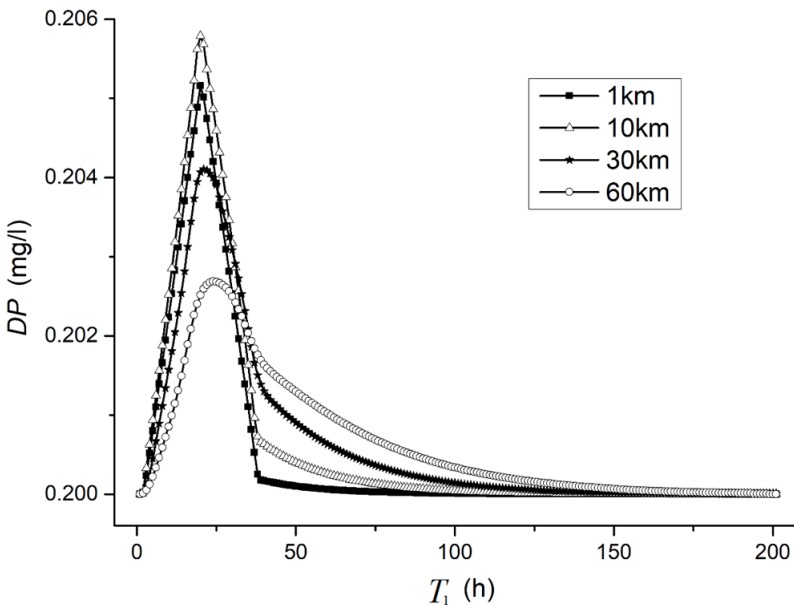

**Figure 8.** The changing process of *DP* concentration of case *AP*-0.04-36 over time.

Because the change in the incoming boundary-suspended sediment will aggravate the sedimentation and resuspension process between the suspended sediment and the bed sediment, it affects the migration and transformation process of the phosphorus between the overlying water layer and bed sediment layer. Figure 9 shows the changing process of *DP* concentration of case *SS*-2-36 at the $T_1$ stage. It can be seen that the *DP* shows a peak and a trough shape over time when the peak value of the incoming *SS* changes. Therefore, the temporal and spatial distribution rules of six variables, specifically *DP*, *AP*, *DPS*, *APS*, *SS* and *BS*, are given in Figure 10. The horizontal axis represents the position of the centerline at different distances from the upstream, and the longitudinal axis represents the time. As can be seen, the dissolved phosphorus *DP* and adsorbed phosphorus *AP* in the overlying water layer go through three processes from the initial value to the trough, then to the peak and then to the initial concentration. The first process can be analyzed from Equation (4) because the increase in the *SS* concentration will make the flux of phosphorus settling down larger than that of the bed sediment resuspended upward, resulting in the decrease in *AP*, the desorption of phosphorus into the overlying water and the decrease in *DP*. The increase in the *DPS* in pore water in the bed sediment layer will increase the *DP* through diffusion into the overlying water, and the increase in particulate phosphorus adsorbed by suspended sediment will lead to the increase in *AP*, that is, the second process. After the unsteady process, the incoming suspended sediment returns to the initial value, and the total phosphorus also returns to the initial value, so the third process appears with the *DP* decreases to the initial concentration after reaching the wave crest.

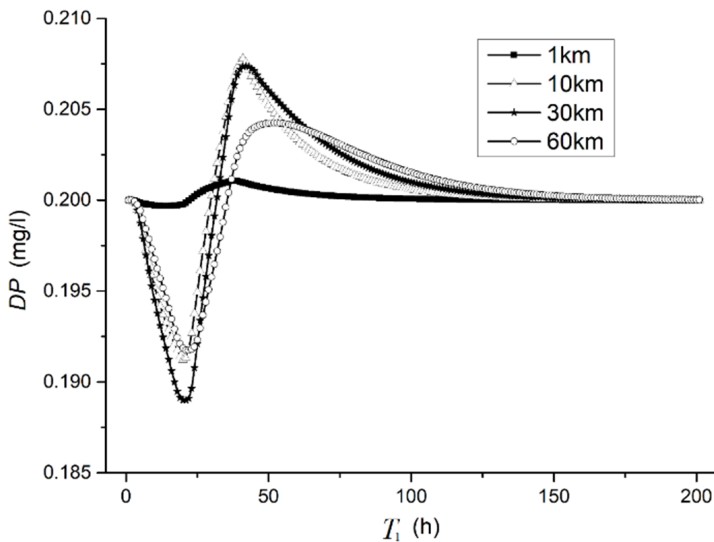

**Figure 9.** The changing process of *DP* concentration of case *SS*-2-36 over time.

**Figure 10.** The spatial-temporal distribution law of (**a**) *DP*, (**b**) *AP*, (**c**) *DPS*, (**d**) *APS*, (**e**) *SS* and (**f**) *BS* of case *SS*-2-36.

The *DPS* and *APS* in the upstream position increases from the initial value to the peak, then decreases to the initial concentration, while the downstream position decreases from the initial value to the trough and then increases to the initial concentration. The reason why there are two opposite processes in the upstream and downstream position is that the suspended sediment from the upper boundary quickly settles in the upstream position, resulting in an increase in phosphorus entering the bed sediment layer. However, the phosphorus flux between the suspended sediment and the bed sediment does not change obviously in the downstream position because the suspended sediment does not increase obviously, but when the upstream *DP* decreases, the convection diffuses to the downstream position, so that the *DP* of this position decreases. Thus the *DPS* decreases, and the above process occurs. The changing trend of the *DP* of other cases, that is, *SS*-0.1/0.25/1/5-36, is consistent with that of Figure 9.

The change in the upstream discharge will either weaken or enhance the upward resuspension process of the bed sediment, thus affecting the migration and transformation process of phosphorus between the overlying water layer and the bed sediment layer. Figure 11 shows the changing process of the *DP* concentration of case *Q*-1000-36 at the $T_1$ stage. It can be seen that when the peak of the incoming discharge changes, the downstream *DP* shows two peaks and one trough over time. Therefore, the temporal and spatial distribution rules of six variables, especially *DP*, *AP*, *DPS*, *APS*, *SS* and *BS*, are given in Figure 12. It is clear that both *DP* and *AP* in the overlying water layer in the upstream region go through three processes: increasing from the initial value to the peak, then decreasing to the trough and then increasing to the initial concentration. The first process can be analyzed from Equations (3)–(8) because the increase in the *SS* concentration of suspended sediment will make the flux of phosphorus settling downwards less than that of the bed sediment resuspended upward, resulting in the increase in *AP* and the increase in phosphorus desorbed into the overlying water, which leads to the increase in *DP*. The increase in *DP* in the overlying water will diffuse into the pore water of the bed sediment layer, resulting in an increase in *DPS* and a decrease in *DP* concentration, so that the phosphorus available for adsorption by suspended sediment decreases, resulting in an increase in *AP*, that is, the second process. After the unsteady process, the incoming flow returns to the initial value, and the sedimentation re-suspension flux returns to the same level, so the third process of *DP* increases to the initial concentration after reaching the trough. In some areas near the downstream, there will be a process from the trough to the secondary peak and then to the initial value, mainly because the velocity in the downstream region is slightly larger than that in the upstream region, so the resuspension rate is slightly larger. As a result, the upward release of phosphorus flux is slightly more than that of sedimentation.

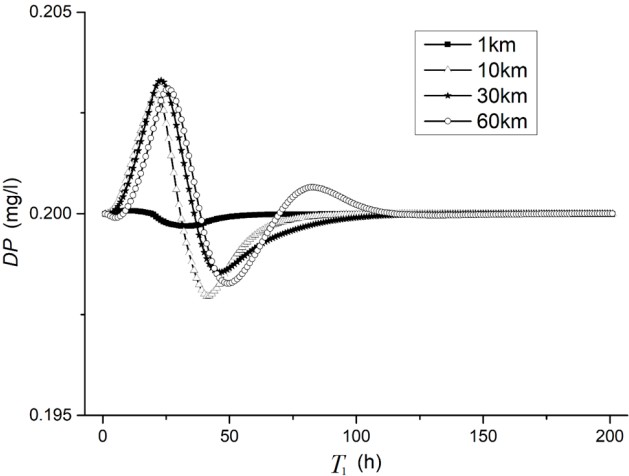

**Figure 11.** The changing process of *DP* concentration of case *Q*-1000-36 over time.

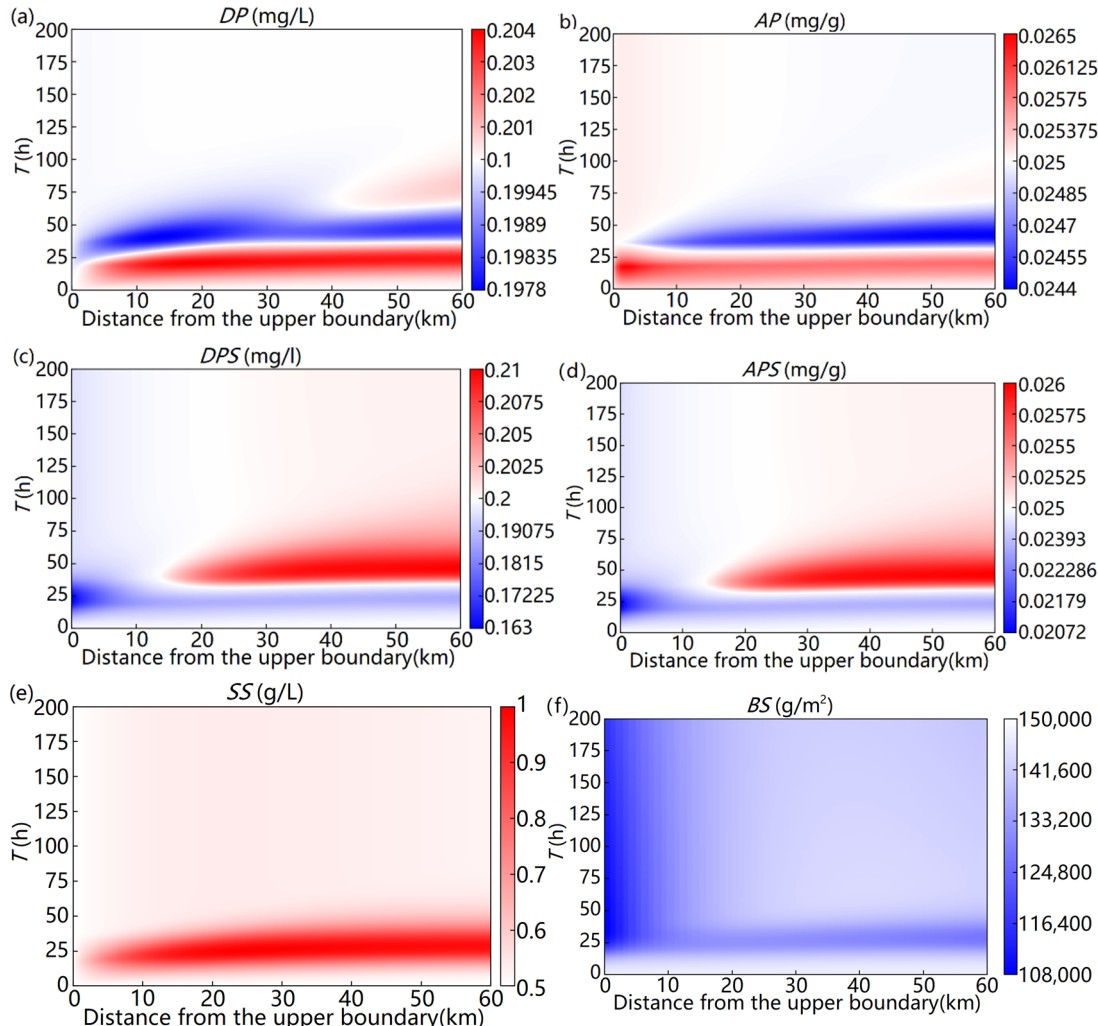

**Figure 12.** The temporal and spatial distribution law of (**a**) *DP*, (**b**) *AP*, (**c**) *DPS*, (**d**) *APS*, (**e**) *SS* and (**f**) *BS* of case *Q-1000-36*.

The *DPS* in pore water and *APS* adsorbed by bed sediment also show different characteristics in the upstream and downstream positions. *DPS* and *APS* in the upstream region will quickly increase from the trough to the initial value due to the action of the *DP* with the same incoming flow boundary. However, the *DP* in the overlying water at the downstream position diffuses into the pore water, resulting in an increase in *DPS* and an increase in phosphorus available for bed sediment adsorption. Accordingly, *APS* increases and finally decreases to nearly the initial value due to the influence of constant inflow.

The maximum *SS* peak concentration will increase with the increase in distance, mainly because the bed sediment will be re-suspended upward along the way, which makes the *SS* increase continuously. The increase in *SS* will increase the bed sediment through sedimentation, so the minimum peak concentration of *BS* appears in the upstream region. In the end, the *BS* in all cross sections along the way will be lower than the initial value, mainly because part of the bed sediment is brought out of the calculation area with the flow. The changing trend of the *DP* of other cases, that is, *Q-350/600/875/1250-36*, is consistent with that of Figure 12.

### 4.2. Maximum Concentration of Dissolved Phosphorus

Through the previous analysis, it was found that the temporal and spatial distribution of phosphorus varied greatly with different types of single factor inflow and different

changes of the same type of inflow, so the peak value of *DP* was selected as the analysis index in this part to analyze its changes along the way (Equation (10)):

$$\Delta DP = \frac{DP_{\max} - DP_{\text{init}}}{DP_{\text{init}}} \tag{10}$$

where $DP_{\max}$ is the peak concentration of *DP* at a certain section, $DP_{\text{init}}$ is the initial concentration of *DP* at a certain section and $\Delta DP$ is the rate of change of *DP*. In order to analyze the changing law of *DP*, 15 points are selected on the centerline of the river, which are 1, 2, 4, 6.5, 10, 15, 20, 25, 30, 35, 40, 45, 50 and 60 km, respectively.

Case *DP*-0.3-36 is selected to analyze the peak concentration of *DP* at each position along the river, as shown in Figure 13. It can be seen that there is a decreasing trend of $DP_{\max}$ along the distance. This is mainly because the suspended sediment and bed sediment will adsorb the dissolved phosphorus. The maximum peak concentration of *DP* appears near the upstream section (1 km), reaching 0.387 mg/L, accounting for 96.75% of the peak value of the incoming flow boundary, indicating that the incoming dissolved phosphorus has a great influence on the distribution of phosphorus along the river.

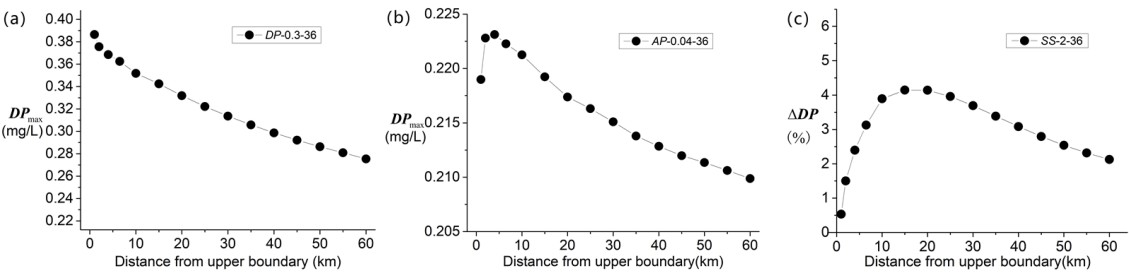

**Figure 13.** The changing law of peak concentration of *DP* along the distance. (**a**) *DP*-0.3-36, (**b**) *AP*-0.04-36, (**c**) *SS*-2-36.

The peak concentrations of *DP* at each position along the river are given by the same method in the cases of *AP*-0.04-36 and *SS*-2-36, respectively, as shown in Figure 13. It can be seen that the two cases no longer show a monotonous decreasing trend with the distance. Because the adsorbed phosphorus *AP* affects the dissolved phosphorus *DP* in the water through sediment desorption, and the action time is obviously longer than the convection diffusion time but the incoming *DP* does not change in this section, so the first several sections are affected by the constant boundary *DP*, which will weaken the effect of *AP* on the *DP* at this position. Therefore, when the incoming flow *AP* changes, the maximum concentration of downstream *DP* does not appear in the first section. The peak concentration of *DP* in *SS*-2-36 shows a trend of increasing at first and then decreasing.

The maximum peak concentration of *DP* of case *AP*-0.04-36 appears in 4 km, reaching 0.223 mg/L, which is 11.5% higher than the initial value, indicating that incoming adsorbed phosphorus has a strong effect on the distribution of phosphorus along the river. The maximum peak concentration of the *DP* of case *SS*-2-36 appears in 15 km, which is only 4.2% higher than the initial value, indicating that the incoming suspended sediment has a weak effect on the distribution of phosphorus along the route.

The effect of inflow discharge on phosphorus is very weak because the maximum change rate of the *DP* peak concentration of all relative cases is only 4.25%, no further analysis is made here.

### 4.3. Influence of Maximum Value of Incoming Flow Variable

Through the above analysis, it was found that there was a great difference in the peak concentration of *DP* downstream with different inflow changes, so this section focused on the comparison of the relationship between the peak concentration change rate and the inflow variable change rate:

$$\Delta DP_{\text{in}} = \frac{DP_{\text{in\_max}} - DP_{\text{in\_init}}}{DP_{\text{in\_init}}}, \ \Delta AP_{\text{in}} = \frac{AP_{\text{in\_max}} - AP_{\text{in\_init}}}{AP_{\text{in\_init}}}, \ \Delta SS_{\text{in}} = \frac{SS_{\text{in\_max}} - SS_{\text{in\_init}}}{SS_{\text{in\_init}}} \quad (11)$$

where $DP_{\text{in\_max}}$ is the peak concentration of $DP$ on the incoming flow boundary, $DP_{\text{in\_init}}$ is the initial concentration of $DP$ on the incoming flow boundary, $\Delta DP_{\text{in}}$ is the change rate of $DP$ on the incoming flow boundary, $AP_{\text{in\_max}}$ is the peak concentration of $AP$ on the incoming flow boundary, $AP_{\text{in\_init}}$ is the initial concentration of $AP$ on the incoming flow boundary, $\Delta AP_{\text{in}}$ is the change rate of $AP$ on the incoming flow boundary, $SS_{\text{in\_max}}$ is the peak concentration of $SS$ on the incoming flow boundary, $SS_{\text{in\_init}}$ is the initial concentration of $SS$ on the incoming flow boundary and $\Delta SS_{\text{in}}$ is the change rate of $SS$ on the incoming flow boundary.

The peak concentration of $DP$ at the 15 points of the 5 cases of $DP$-0.02/0.1/0.3/0.4/1-36 and the peak value of the incoming $DP$ are shown in Figure 14. According to the analysis along the longitudinal axis, the absolute value of the change rate of $DP$ peak value of each case decreases with the increase in distance, and the change rate of the $DP$ peak value at the position of 1 km is 2 times larger than that of 60 km from the upper boundary in each case, mainly because the upstream is strongly affected by the incoming flow. According to the analysis along the transverse direction, the curve of the relationship between the peak change rate of $DP$ at different positions and the peak change rate of incoming flow can be approximately regarded as a straight line passing through the origin, but the slope decreases with the increase in distance, and the slopes at 1 km and 60 km are 0.936 and 0.403, respectively, which means most of the peak $DP$ at the downstream boundary has been reduced after sediment adsorption along the river.

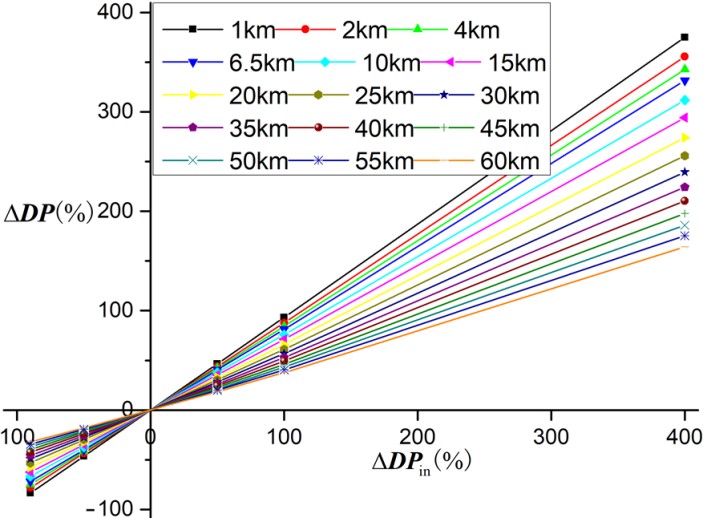

**Figure 14.** The relationship between the peak value of $DP$ along the river and the incoming $DP$ of case $DP$-0.02/0.1/0.3/0.4/1-36.

The slope of each curve in Figure 14, that is, the relationship between the change rate of the peak concentration of $DP$ along the river and the incoming $DP$, is shown in Equation (11). According to the statistics of the coefficient shown in Figure 15, there is an exponential decreasing trend with the distance, that is, the rate decreases faster near the upstream section, and it decreases slowly with the increase in distance. The main reason is that the suspended sediment and bed sediment will absorb dissolved phosphorus or release adsorbed phosphorus into the water body, and the closer to the upstream, the greater the concentration difference, and the more obvious the sediment adsorption and desorption,

the greater the decreasing rate. With the increase in distance, the concentration difference becomes smaller and smaller, so the decreasing rate is also smaller:

$$\Delta DP = \alpha \cdot \Delta DP_{\text{in}}, \Delta DP = \beta \cdot \Delta AP_{\text{in}}, \Delta DP = \gamma \cdot \Delta SS_{\text{in}} \tag{12}$$

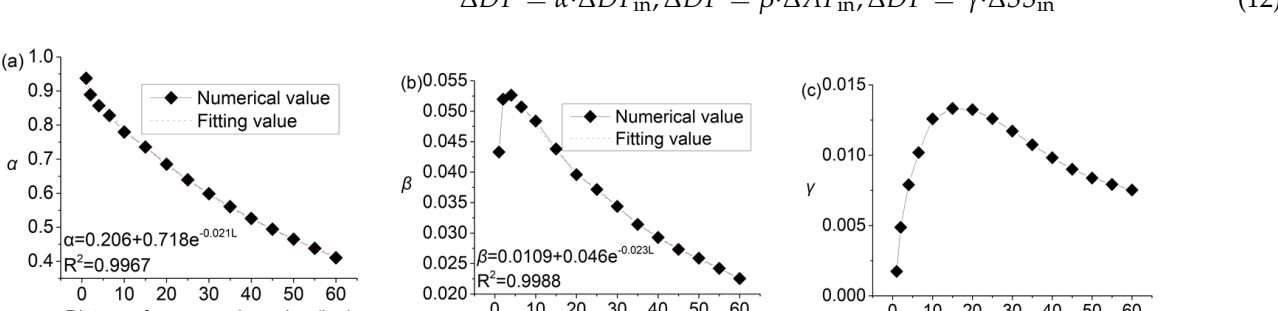

**Figure 15.** The relationship between the change rate of the peak concentration of *DP* along the river and the incoming (**a**) *DP*, (**b**) *AP* and (**c**) *SS*.

The same method is used to analyze the relationship between the peak concentration of *DP* and the peak value of incoming *AP* at 15 points of case *AP*-0.005/0.01/0.04/0.08/0.12-36. The curve can be approximately regarded as a straight line passing through the origin, and the slope $\beta$ is shown in Equation (12).

Compared with $\alpha$, $\beta$ greatly decreases. The maximum value of $\beta$, 0.0526, appears at the 4 km from the upstream, that is, when the peak of the incoming *AP* increases by 100%, the *DP* peak value at the 4 km position of the cross section increases by 5.26%. The first three sections are gradually increasing. There is an exponential decreasing trend along the river, that is, the rate decreases rapidly near the upstream section, and it decreases more and more slowly with the increase in the distance. The main reason is that the adsorbed phosphorus will be released into the water body, and the closer it is to the upstream, the greater the concentration difference, and the more obvious the sediment adsorption and desorption, the greater the decreasing rate. With the increase in distance, the concentration difference is smaller, so the reduction rate is smaller.

The temporal and spatial distribution law of *DP*, *AP*, *DPS*, *APS*, *SS* and *BS* of case *SS*-0.1/0.25-36 is opposite to that in Figure 10. The law of another two cases is basically the same as that in Figure 10, but there are some differences in the peak values of different cases. Since the focus of the analysis is on the positive influence of the inflow conditions on the downstream, the peak value of the second process of *DP* is taken as the research object. According to the relationship between the peak concentration of *DP* and the peak value of incoming *SS*, the curve of the relationship between the peak concentration of *DP* and the peak change rate of incoming *SS* at different locations along the river can be approximately regarded as a straight line passing through the origin, and the slope $\gamma$ is shown in Equation (12).

Compared with $\alpha$ and $\beta$, $\gamma$ greatly decreases. The maximum value, 0.0132, appears at the 20 km from the upstream, that is, when the peak value of incoming *SS* increases by 100%, the peak value of *DP* increases by 1.32%. As a whole, it shows a trend of first increasing and then decreasing, which is basically consistent with the peak law of *DP* during the change of incoming suspended sediment.

### 4.4. Influence of Total Phosphorus of Incoming Flow

In the previous analysis, there is a significant difference between $\alpha$, $\beta$ and $\gamma$, partly because of the great difference in total phosphorus TP caused by different variables. Therefore, this part intended to explore whether there was a direct relationship between the change rate of total incoming phosphorus and the change rate of *DP* peak concentration (Equations (13) and (14)):

$$TP = DP + AP \cdot SS \tag{13}$$

$$\Delta TP_{\text{in}} = \frac{TP_{\text{in\_max}} - TP_{\text{in\_init}}}{TP_{\text{in\_init}}} \tag{14}$$

where $TP_{\text{in\_max}}$ is the peak concentration of incoming $TP$, $TP_{\text{in\_init}}$ is the initial concentration of incoming $TP$ and $\Delta TP_{\text{in}}$ is the rate of change of incoming $TP$.

The maximum change rate of the $DP$ peak concentration is selected as the research object. Table 3 shows the relationship between $\Delta DP_{\text{max}}$ and $\Delta TP_{\text{in}}$ with the change in the incoming $DP$ or $AP$. Figure 16 shows the relationship between the change rate of $DP$ and the rate of incoming $TP$ with the cases that incoming $DP$ or $AP$ changes, and a straight line with a slope of 0.9951 is obtained, whose $R^2$ is 0.9999, indicating that there is a direct relationship between the distribution of $DP$ along the river and the total phosphorus $TP$, that is, when the $TP$ of the incoming flow changes, the $DP$ along the downstream will almost have the same rate of change as the initial value.

**Table 3.** The relationship between $\Delta DP_{\text{max}}$ and $\Delta TP_{\text{in}}$ with the change of incoming $DP$ or $AP$.

| Case | $DP_{\text{in\_max}}$ (mg/L)/ $AP_{\text{in\_max}}$ (mg/g) | $TP_{\text{in\_max}}$ (mg/L) | $\Delta DP_{\text{in}}$/ $\Delta AP_{\text{in}}$ (%) | $\Delta TP_{\text{in}}$ (%) | $\Delta DP_{\text{max}}$ (%) |
|------|------|------|------|------|------|
| *DP*-0.02-36 | 0.02 | 0.0325 | −90.0 | −84.7 | −83.6 |
| *DP*-0.1-36 | 0.1 | 0.1125 | −50.0 | −47.1 | −46.5 |
| *DP*-0.3-36 | 0.3 | 0.3125 | 50.0 | 47.1 | 46.6 |
| *DP*-0.4-36 | 0.4 | 0.4125 | 100.0 | 94.1 | 93.2 |
| *DP*-1-36 | 1 | 1.0125 | 400.0 | 376.5 | 374.9 |
| *AP*-0.005-36 | 0.005 | 0.2025 | −79.2 | −4.7 | −4.2 |
| *AP*-0.01-36 | 0.01 | 0.205 | −60.0 | −3.5 | −3.1 |
| *AP*-0.04-36 | 0.04 | 0.22 | 60.0 | 3.5 | 3.2 |
| *AP*-0.08-36 | 0.08 | 0.24 | 220.0 | 12.9 | 11.6 |
| *AP*-0.12-36 | 0.12 | 0.26 | 380.0 | 22.4 | 20.0 |

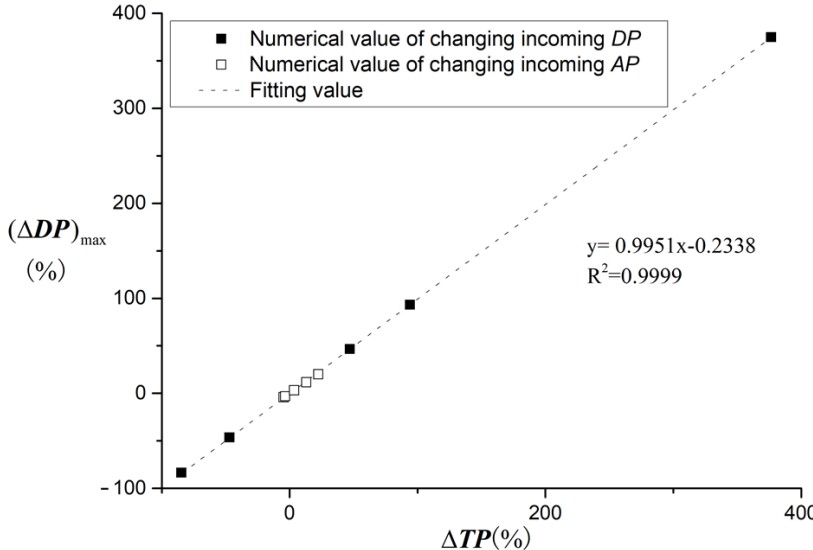

**Figure 16.** The relationship between the changing rate of *DP* and the rate of incoming TP with the cases that incoming *DP* or *AP* changes.

Table 4 shows relationship between $\Delta DP_{\text{max}}$ and $\Delta TP_{\text{in}}$ with the change in the incoming $SS$. Figure 17 shows the relationship between the change rate of the $DP$ and the rate of incoming $TP$ with the cases that the incoming $AP$ or $SS$ changes, and a straight line with a slope of 0.2154 is obtained whose $R^2$ is 0.9955, indicating that when the incoming suspended sediment changes, the distribution of the $DP$ along the river has a strong relationship with total phosphorus $TP$, that is, when the change in the incoming flow $TP$ is 100%, the maximum peak concentration of the $DP$ along the downstream is 21.54% higher

than that of the initial. The reason why the change in the incoming flow *SS* has less influence on the temporal and spatial distribution of downstream *DP* than that of the incoming flow *DP* or *AP* is that the change of *SS* will aggravate the process of sedimentation and re-suspension between suspended sediment and bed sediment, thus affecting the migration and transformation process of phosphorus between the overlying water layer and bed sediment layer, and the increase in sediment will also reduce the phosphorus adsorption per unit sediment.

**Table 4.** The relationship between $\Delta DP_{max}$ and $\Delta TP_{in}$ with the change in incoming *SS*.

| Case | $SS_{in\_max}$ (g/L) | $TP_{in\_max}$ (mg/L) | $\Delta SS_{in}$ (%) | $\Delta TP_{in}$ (%) | $\Delta DP_{max}$ (%) |
|---|---|---|---|---|---|
| *SS*-0.1-36 | 0.1 | 0.2025 | −80.0 | −4.7 | −0.1 |
| *SS*-0.25-36 | 0.25 | 0.20625 | −50.0 | −2.9 | −0.1 |
| *SS*-1-36 | 1 | 0.225 | 100.0 | 5.9 | 1.4 |
| *SS*-2-36 | 2 | 0.25 | 300.0 | 17.7 | 4.1 |
| *SS*-5-36 | 5 | 0.325 | 900.0 | 52.9 | 12.0 |

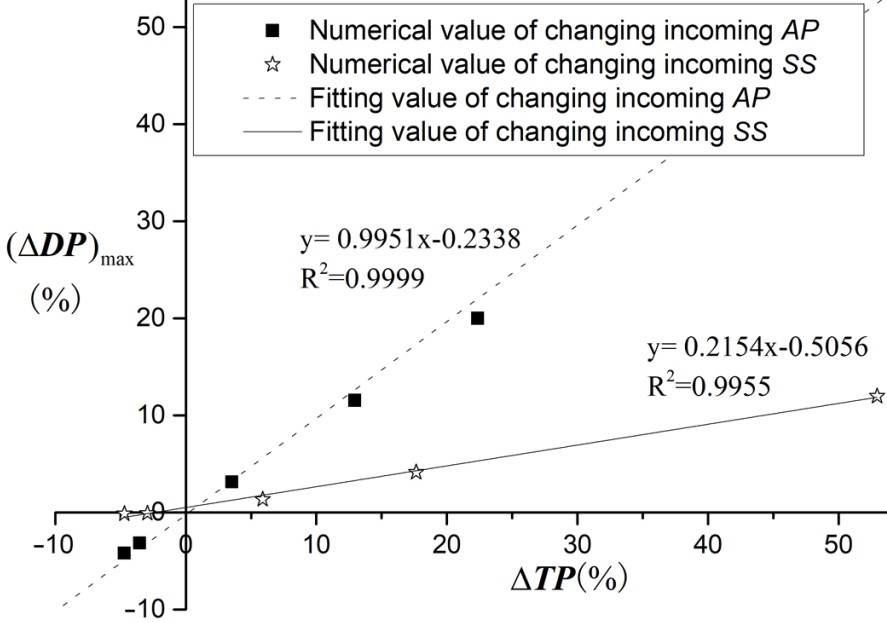

**Figure 17.** The relationship between the changing rate of *DP* and the rate of incoming TP with the cases that incoming *AP* or *SS* changes.

*4.5. Comparison between Maximum Value and Total Phosphorus of Incoming Flow*

This part intends to compare and analyze the effects of not-constant time and peak concentration of different inflow variables. The $\Delta DP_{max}$ of nine groups of each kind of cases are shown in Figure 18 in the sequence of not-constant time and peak concentration. Taking the case that *DP* changes, the numbers 1~9 in Figure 18a are cases *DP*-0.02-36, *DP*-0.1-36, *DP*-0.3-36, *DP*-0.4-12, *DP*-0.4-24, *DP*-0.4-36, *DP*-0.4-48, *DP*-0.4-60 and *DP*-1-36, respectively.

Figure 18a shows the maximum change rate of the *DP* peak concentration with nine groups of each case where the incoming *DP* changes. It can be seen that when the not-constant time is fixed at 36 h and the peak *DP* of the incoming flow is 0.02, 0.1, 0.3, 0.4 and 0.5 mg/L, the maximum change rate of the peak *DP* along the river is −31.99, 18.00, 46.57, 93.23 and 374.91%, respectively, and the change is very obvious. However, the peak *DP* of the incoming flow is fixed as 0.4 mg/L, and the not-constant time is 12, 24, 36, 48 and 60 h, the maximum change rate of the *DP* peak along the river is 86.61, 91.48, 93.23, 94.16 and 94.75%, respectively, and the maximum difference among the five cases is 8.6%, indicating

that the influence of the not-constant time of the incoming flow on the downstream *DP* peak is weak.

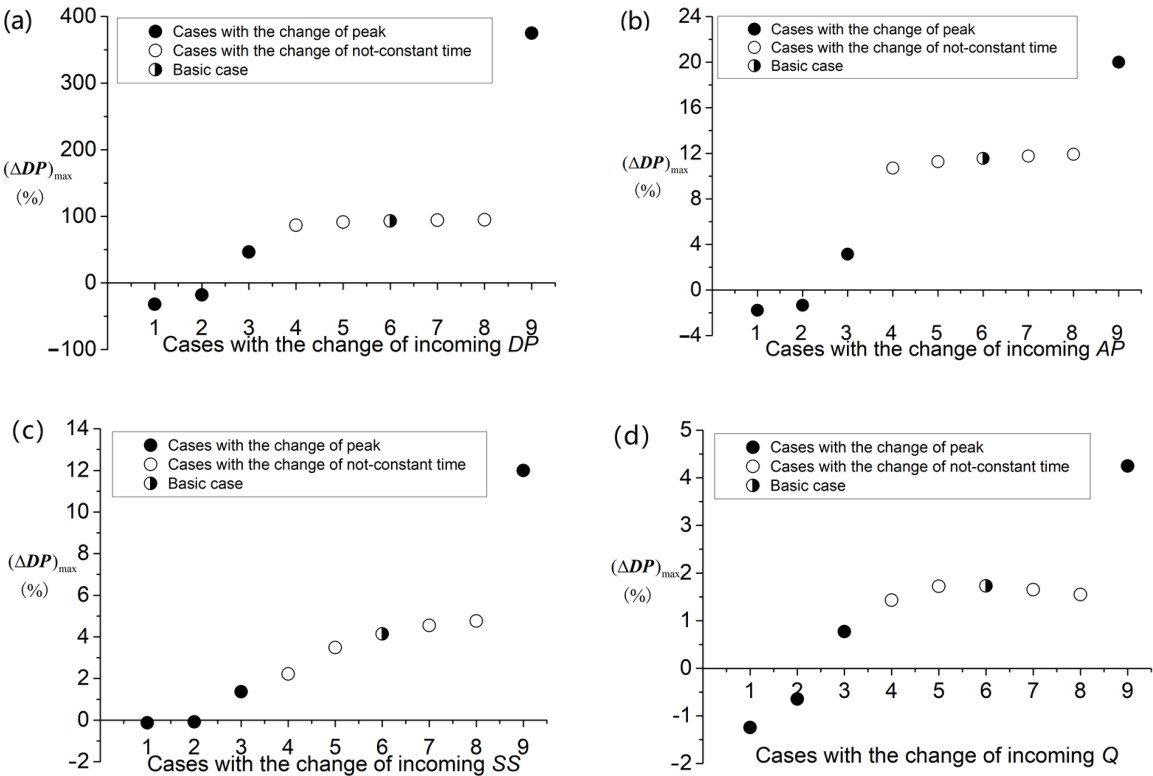

**Figure 18.** (**a**–**d**) The maximum change rate of downstream *DP* peak with the case that incoming (**a**) *DP*/(**b**) *AP*/(**c**) *SS*/(**d**) *Q* changes.

Figure 18b shows the maximum change rate of *DP* peak with the change of incoming *AP*. It can be seen that when the not-constant time is fixed at 36 h and the peak *AP* of the incoming flow is 0.005, 0.01, 0.04, 0.08, 0.1 mg/g, the maximum change rates of *DP* along the river are −1.79, 1.34, 3.15, 11.56 and 19.99%, respectively. However, the peak *AP* of the incoming flow is fixed as 0.08 mg/g, and when the not-constant time is 12, 24, 36, 48, 60 h, the maximum change rate of the *DP* peak along the river is 10.71, 11.27, 11.56, 11.76 and 11.92%, respectively, and the maximum difference between the five groups is 10.1%, indicating that the influence of the not-constant time of the incoming flow on the downstream *DP* peak is weak which is consistent with the change in the *DP* in the incoming flow.

Figure 18c shows the maximum change rate of the *DP* peak with the change of incoming *SS*. It can be seen that when the not-constant time is fixed at 36 h and the peak *SS* of the incoming flow is 0.1, 0.25, 1, 2, 5 g/L, the maximum change rates of *DP* along the river are 0.13, 0.08, 1.36, 4.14 and 11.99%, respectively. The peak *SS* of the incoming flow is fixed as 2 g/L, and when the not-constant time is 12, 24, 36, 48 and 60 h, the maximum change rate of the *DP* peak along the river is 2.22, 3.49, 4.14, 4.55 and 4.77%, respectively, and the maximum difference among the five groups is 52.4%, indicating that the effect of the not-constant time of the incoming flow on the downstream *DP* peak is equivalent to the change in the incoming flow peak.

Figure 18d shows the maximum change rate of *DP* peak concentration with the change in the incoming *Q*. It can be seen that $\Delta DP_{max}$ is only 4.25%, indicating that the impact of discharge changes on the peak value of *DP* is very weak, and there is no obvious "secondary pollution" due to large flow disturbance. The reason why there is no obvious "secondary pollution" phenomenon may be that the dissolved phosphorus *DPS*

and adsorbed phosphorus *APS* in the sediment are the same as the adsorbed phosphorus *DP* and particulate phosphorus *AP* in the overlying water, respectively. In previous studies, the phenomenon of "secondary pollution" is that the pollution degree of the bottom is much higher than that of the incoming water quality, and the large flow disturbance will enhance the upward suspension of the bottom bed sediment and significantly increase the upward release of phosphorus flux, resulting in the increase in total phosphorus in the overlying water, namely, the phenomenon of "secondary pollution".

## 5. Conclusions

In this paper, based on the established equilibrium condition of the generalized river, the unsteady process was set as different peak concentration and time to explore the relationship between the unsteady process and the spatial-temporal distribution law of phosphorus along the river by separately changing the dissolved phosphorus, adsorbed phosphorus, suspended sediment and discharge. The main conclusions are as follows:

(1) Under the action of different incoming variables, the distribution law of downstream phosphorus is quite different. When *DP* or *AP* changes, the overall trend of the downstream *DP* is consistent with that of the incoming *DP/AP*, that is, it increases or decreases with the increase or decrease in the incoming *DP/AP*. When *SS* changes, it will aggravate the sedimentation and resuspension process between the suspended sediment and the bed sediment and affect the migration and transformation process of phosphorus between the overlying water layer and the bed sand layer. Thus, when the suspended sediment concentration increases, the *DP* and *AP* will go through three processes: first decreasing from the initial value to the wave trough, then increasing to the wave peak and then decreasing to the initial concentration. When *Q* changes, it will change the resuspension rate of bed sediment, thus affecting the distribution of phosphorus. Thus, when the discharge increases, the *DP* and *AP* in most areas will go through three processes: first increasing from the initial value to the peak, then decreasing to the trough, and then increasing to the initial concentration.

(2) The effects of the four variables on the peak value of downstream *DP* are quite different. When incoming *DP* changes, $DP_{\max}$ decreases with the distance, while when incoming *AP* or *SS* changes, $DP_{\max}$ increases at first and then decreases with the distance, and the maximum value appears in the 4 and 15 km cross-section, respectively. The maximum values of $DP_{\max}$ with the change in *DP*, *AP* and *SS* are 96.75%, 11.5% and 4.2%, respectively, but those with the change in *Q* is very small.

(3) The relationship curve between $DP_{\max}$ and the change rate of the incoming flow variables at different positions along the river can be approximately regarded as a straight line passing through the origin, and these slopes are recorded as $\alpha$, $\beta$ and $\gamma$, respectively. $\alpha$ decreases along the river; $\beta$ increases along the first three sections and then decreases; and $\gamma$ shows a trend of first increasing and then decreasing, and the maximum value appears at the 20 km cross-section, which is only 0.0132, much less than $\alpha$ and $\beta$.

(4) When incoming *DP* or *AP* changes, the maximum change rate between $\Delta DP_{\max}$ and $\Delta TP_{\mathrm{in}}$ is fitted into a straight line with a slope of 0.9951, while when the incoming *SS* changes, there is a straight line with a slope of 0.2154.

(5) The effect of the not-constant time of the inflow on the peak concentration of phosphorus along the river is much weaker than that of the peak concentration of the incoming flow. When the peak concentration of *DP* and *AP* changes, $\Delta DP_{\max}$ changes greatly, but when the not-constant time changes, it is very small, and the maximum differences of the five groups of each case are only 8.6 and 10.1%, respectively. When the not-constant time or peak concentration of incoming *SS* changes, $\Delta DP_{\max}$ is more obvious, mainly based on the small value. The $\Delta DP_{\max}$ is only 4.25% with the change of incoming *Q*, indicating that the effect of discharge on the peak value of *DP* along the river is very weak.

**Author Contributions:** P.H. built and verified the numerical model, discussed the results and participated in the writing. J.X. and L.W. carried out the analysis of the methodology and participated in the writing. The macroscopic idea of the paper is given by H.T. and M.W.; P.X. checked the errors of chart and syntax. All authors have read and agreed to the published version of the manuscript.

**Funding:** This research is funded by the National Key R&D Program of China(2021YFC3200403), the National Natural Science Foundation of China (51879086), the Fundamental Research Funds for the Central Universities(B200204044), the 111 Project (B17015) and Excellent Scientific and Technological Innovation Team in Jiangsu Province.

**Acknowledgments:** The authors would like to acknowledge the support of the Sediment lab at Hohai University.

**Conflicts of Interest:** The authors declare no conflict of interest.

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
