# Peer review of "Influence of Different Single Factors on the Spatial-Temporal Distribution Law of Phosphorus in the Generalized River"

_sustainability, doi:10.3390/su14042070_

Round 1

Reviewer 1 Report

1. General comment:
Interesting analytical approach (including phase-distribution kinetic models) to describe and predict phosphorus distribution between dissolved/adsorbed and deposited in a aqueous body (a generalized river), based in simple physical experiments.
Presented data are "theoretical" (data simulations in a generalized river) however these simulations can help in predicting and describing phosphorus distribution profiles in time and distance from contamination source.
It will be a serious improvement to have experimental results (based on a actual given river) or laboratorial (small-scale lab tank).
Other several factors may also be very important, conditioning distribution profiles such as medium pH, temperature, ionic strength and respective inorganic composition (phosphorus precipitates with several di-valent and three-valent  cations). 

2. Suggestions:
S1: in line 97 correct "10-6m2/s" to "10⁻⁶ m².s⁻¹"

S2: in Tables 3 and 4
> when reporting estimated percentages (%), please keep a maximum of 1 decimal place
(we are dealing with simulated values)

S3: please check your references again (complete some missing information)

Author Response

Dear reviewer 1,

Thank you for your kind comments on our manuscript entitled “Influence of Different Single Factors on the Spatial-Temporal Distribution Law of Phosphorus in the Generalized River”. Thank you very much for giving us an opportunity to revise our manuscript. We revised the manuscript in accordance with comments, and carefully proof-read the manuscript to minimize typographical, grammatical, and bibliographical errors.

A revised manuscript by using the “Track Changes” function has been submitted for easy checking/editing purpose. We hope this revision can make our paper more acceptable. If you have any questions, please contact us without hesitation. The responses to the comments are addressed point by point below.

  1. General comment
    Interesting analytical approach (including phase-distribution kinetic models) to describe and predict phosphorus distribution between dissolved/adsorbed and deposited in an aqueous body (a generalized river), based in simple physical experiments.
    Presented data are "theoretical" (data simulations in a generalized river) however these simulations can help in predicting and describing phosphorus distribution profiles in time and distance from contamination source.
    It will be a serious improvement to have experimental results (based on an actual given river) or laboratorial (small-scale lab tank).
    Other several factors may also be very important, conditioning distribution profiles such as medium pH, temperature, ionic strength and respective inorganic composition (phosphorus precipitates with several di-valent and three-valent cations). 

Response to 1 General comment: Thank you for your approval of the research methods and conclusions of this paper. Field experiments and physical flume experiments do improve this paper greatly. Actually, we have carried out a six-day field experiment based on Bengbu Reach of Huaihe River. The results of this part are being summarized and ready to be submitted to the corresponding journals. Many references show that pH, temperature and ionic strength contribute to the distribution of phosphorus. In the future, I will study the impact of these factors on the distribution of phosphorus to improve the water quality model which can more accurately reflect the laws of real rivers.

  1. Suggestions
    S1: in line 97 correct "10-6m2/s" to "10⁻⁶ m².s⁻¹"

Response to S1: We take the suggestion and revised them. Moreover, we have carefully checked the rest of the superscripts and subscripts. In Table, g/m2 has been changed to g/m2.

S2: in Tables 3 and 4
> when reporting estimated percentages (%), please keep a maximum of 1 decimal place
(we are dealing with simulated values)

Response to S2: We take the suggestion and has kept a maximum of 1 decimal place of all the estimated percentages in Table 3 and 4.

S3: please check your references again (complete some missing information)

Response to S3: We have carefully checked all the references, and corrected 2/11/16/23/24/25/26/27 which are listed as below. Moreover, an additional recent literature reference was added with the number 17.

  1. Conley, D.J.; Paerl, H.W.; Howarth, R.W.; Boesch, D.F.; Seitzinger, S.P. Controlling eutrophication:nitrogen and phosphorus. Science 2009, 323(5917), 1014-1015.
  2. Lopez, P.; Lluch, X.; Vidal, M.; Morgui, J.A. Adsorption of phosphorus on sediments of the Balearic Islands (spain) related to their composition. Estuarine Coastal & Shelf Science 1996, 42(2), 185-196.
  3. Rigler, F.H. A dynamic view of the phosphorus cycle in lakes. Environmental Phosphorus Handbook 1973.
  4. Taylor, M.; Akbarzadeh, Z.; Cappellen, P.V. Global dam-driven changes to riverine N:P:Si ratios delivered to the coastal ocean. Geophysical Research Letters 2020, 47(15), e2020GL088288.
  5. Chen, L.G.; Shi, Y.; Qian, X.; Jin, Q.; Lai, X.Z.; Wang, S. Coupled mathematical model of hydrology-hydrodynamics-water quality in gate-controlled river network-Ⅱ.Application. Advances in Water Science 2014, 25(6), 856-863.
  6. Chen, L.G.; Shi, Y.; Qian, X.; Luan, Z.Y.; Jin, Q. Coupled mathematical model of hydrology -hydrodynamics-water quality in gate-controlled river network-Ⅰ.Theory. Advances in Water Science 2014, 25(4), 534-541.
  7. Hu, P.J.; Wang, L.L.; Li, Z.W.; Zhu, H.; Tang, H.W. Numerical Simulation of the Interaction between Phosphorus and Sediment Based on the Modified Langmuir Equation. Water 2018, 10, 840.
  8. Li, Z.W.; Tang, H.W.; Xiao, Y.; Zhao, H.Q.; Li, Q.X.; Ji, F. Factors influencing phosphorus adsorption onto sediment in a dynamic environment. Journal of Hydro-environment Research 2016, 10, 1-11.

  1. Wu, P.; Wang, N.R.; Zhu. L.J.; Lu, Y.J.; Fan, H.X.; Lu, Y. Spatial-temporal distribution of sediment phosphorus with sediment transport in the Three Gorges Reservoir. Science of the Total Environment2021, 769(4), 144986.

Reviewer 2 Report

This manuscript titled as “Influence of Different Single Factors on the Spatio-Temporal Distribution Law of Phosphorus in the Generalized River”. Overall,   the manuscript is well written, structured and it represents the interesting information regarding the Single Factors on the Spatio-Temporal Distribution Law of Phosphorus using a numerical model of three-dimensional periodic flume, and verified it. However, in my opinion the manuscript has some part that should be improved. I strongly recommend the paper for acceptance after the following correction.

  1. The grammar used in the manuscript must be improved.
  2. Abbreviation must be given in appropriate places (First appearance not later-Page number 3 line number 104-113).
  3. Typo errors should be reduced.
  4. The letters inside the figures are not clearly visible (Figure-1)
  5. Follow same pattern to prepare the figure. In some places closed brackets missing in the data labels
  6. In Figure there is no space between the axis title and their unit
  7. In Figure 3. Explain the anomalous behavior at higher flow velocity with water depth
  8. 4 not clear, please give the values neatly and legibly
  9. In Figure 10. Remove the typo error in the figure captions.
  10. In Figure 12 and 13 is not visible. Kindly change it.
  11. Page number 13 equation is given in bold others equations are in normal text.
  12. Recheck the Figure 15. Figure details is missing in third figure
  13. Recheck the Figure 17 labels
  14. Give an additional recent literature reference.

Author Response

Dear reviewer 2,

Thank you for kind comments on our manuscript entitled “Influence of Different Single Factors on the Spatio-Temporal Distribution Law of Phosphorus in the Generalized River”. Thank you very much for giving us an opportunity to revise our manuscript. We revised the manuscript in accordance with comments, and carefully proof-read the manuscript to minimize typographical, grammatical, and bibliographical errors.

A revised manuscript by using the “Track Changes” function has been submitted for easy checking/editing purpose. We hope this revision can make our paper more acceptable. If you have any questions, please contact us without hesitation. The responses to the comments are addressed point by point below.

  1. The grammar used in the manuscript must be improved.

Response to 1: We have carefully checked the grammatical problems in this article, compared the three tenses of the general present tense, the past tense and the present perfect tense, and corrected the ones that should be used in the past tense. Taking corrected “Abstract” as an example which is shown as below.

  1. Abbreviation must be given in appropriate places (First appearance not later-Page number 3 line number 104-113).

Response to 2: It was found that all the abbreviations which appeared in Figure 1 had been explained in the line 104-113, and the rest of the abbreviations had been explained when they first appeared.

  1. Typo errors should be reduced.

Response to 3: We have carefully checked the typo errors, and found there were several errors which are listed as below.

(1)Spatio-temporal~spatial-temporal

(2)10-6m2/s~10⁻⁶ m²/s

(3) g/m2~g/m2

(4)"Figure 7.The changing..."~"Figure 7. The changing... "

(5)Reference: 2, 11, 16, 23, 24, 25, 26, 27

  1. Conley, D.J.; Paerl, H.W.; Howarth, R.W.; Boesch, D.F.; Seitzinger, S.P. Controlling eutrophication:nitrogen and phosphorus. Science 2009, 323(5917), 1014-1015.
  2. Lopez, P.; Lluch, X.; Vidal, M.; Morgui, J.A. Adsorption of phosphorus on sediments of the Balearic Islands (spain) related to their composition. Estuarine Coastal & Shelf Science 1996, 42(2), 185-196.
  3. Rigler, F.H. A dynamic view of the phosphorus cycle in lakes. Environmental Phosphorus Handbook 1973.
  4. Taylor, M.; Akbarzadeh, Z.; Cappellen, P.V. Global dam-driven changes to riverine N:P:Si ratios delivered to the coastal ocean. Geophysical Research Letters 2020, 47(15), e2020GL088288.
  5. Chen, L.G.; Shi, Y.; Qian, X.; Jin, Q.; Lai, X.Z.; Wang, S. Coupled mathematical model of hydrology-hydrodynamics-water quality in gate-controlled river network-Ⅱ.Application. Advances in Water Science 2014, 25(6), 856-863.
  6. Chen, L.G.; Shi, Y.; Qian, X.; Luan, Z.Y.; Jin, Q. Coupled mathematical model of hydrology -hydrodynamics-water quality in gate-controlled river network-Ⅰ.Theory. Advances in Water Science 2014, 25(4), 534-541.
  7. Hu, P.J.; Wang, L.L.; Li, Z.W.; Zhu, H.; Tang, H.W. Numerical Simulation of the Interaction between Phosphorus and Sediment Based on the Modified Langmuir Equation. Water 2018, 10, 840.
  8. Li, Z.W.; Tang, H.W.; Xiao, Y.; Zhao, H.Q.; Li, Q.X.; Ji, F. Factors influencing phosphorus adsorption onto sediment in a dynamic environment. Journal of Hydro-environment Research 2016, 10, 1-11.

  1. The letters inside the figures are not clearly visible (Figure-1)

Response to 4: We changed the corresponding letters and made a higher-precision picture shown as below.

  1. Follow same pattern to prepare the figure. In some places closed brackets missing in the data labels

Response to 5: Most pictures were made by ORIGIN and PYTHON. We have found that missing brackets and other errors were because of not selecting the setting “Do not compress the image in the file” in the WORD. After selecting it, all the missing or ambiguous problems can be solved.

  1. In Figure there is no space between the axis title and their unit

Response to 6: We have corrected all the problems. Taking corrected “Figure 8” as an example which is shown as below.

  1. In Figure 3. Explain the anomalous behavior at higher flow velocity with water depth

Response to 7: The veloctity with water depth follows the logarithmic law as a whole, but the maximum velocity does’t appear in the water surface due to the action of sediment. Zhaoying Wang* concluded that the maximum velocity in open channel flow appeared below the surface of the water when the ratio of width to depth is less than 5.2.

*Wang, X.K.; Wang, Z.Y.; Yu, M.Z.; Li, D.X. Velocity profile of sediment suspensions and comparison of log-law and wake-law. Journal of Hydraulic Research 2001, 39(2), 211-217.

  1. Figure 4 not clear, please give the values neatly and legibly

Response to 8: The reason resulting to the problem is the same as question 5.

  1. In Figure 10. Remove the typo error in the figure captions.

Response to 9: We have changed "temporal and spatial” to “spatial-temporal” in the text.

  1. In Figure 12 and 13 is not visible. Kindly change it.

Response to 10: The reason resulting to the problem is the same as question 5.

  1. Page number 13 equation is given in bold others equations are in normal text.

Response to 11: We have changed all variables to be normal (not in bold) in relavent equaitons and the text. Taking corrected “Equation 11” as an example which is shown as below.

  1. Recheck the Figure 15. Figure details is missing in third figure

Response to 12: The reason resulting to the problem is the same as question 5.

  1. Recheck the Figure 17 labels

Response to 13: The reason resulting to the problem is the same as question 5.

  1. Give an additional recent literature reference.

Response to 14: A recent literature was added with number 17.

  1. Wu, P.; Wang, N.R.; Zhu. L.J.; Lu, Y.J.; Fan, H.X.; Lu, Y. Spatial-temporal distribution of sediment phosphorus with sediment transport in the Three Gorges Reservoir. Science of the Total Environment 2021, 769(4), 144986.
